# Explainable Claim Verification via Knowledge-Grounded Reasoning with Large Language Models

**Haoran Wang    Kai Shu**

Illinois Institute of Technology, Chicago, IL, USA

`hwang219@hawk.iit.edu`

`kshu@iit.edu`

## Abstract

Claim verification plays a crucial role in combating misinformation. While existing works on claim verification have shown promising results, a crucial piece of the puzzle that remains unsolved is to understand how to verify claims without relying on human-annotated data, which is expensive to create at a large scale. Additionally, it is important for models to provide comprehensive explanations that can justify their decisions and assist human fact-checkers. This paper presents *First-Order-Logic-Guided Knowledge-Grounded (FOLK) Reasoning* that can verify complex claims and generate explanations without the need for annotated evidence using Large Language Models (LLMs). FOLK leverages the in-context learning ability of LLMs to translate the claim into a First-Order-Logic (FOL) clause consisting of predicates, each corresponding to a sub-claim that needs to be verified. Then, FOLK performs FOL-Guided reasoning over a set of knowledge-grounded question-and-answer pairs to make veracity predictions and generate explanations to justify its decision-making process. This process makes our model highly explanatory, providing clear explanations of its reasoning process in human-readable form. Our experiment results indicate that FOLK outperforms strong baselines on three datasets encompassing various claim verification challenges. Our code and data are available. [1]

---

**Claim:** Lubabalo Kondlo won a silver medal in the 2012 SportAccord World Mind Games inaugurated in July 2011 in Beijing.

**Label:** *[NOT_SUPPORTED]*

**Predicates:**
Won(Lubabalo Kondlo, a silver medal) ::: Verify Lubabalo Kondlo won a silver medal
Inaugurated(the 2012 SportAccord World Mind Games, July 2011, Beijing) ::: Verify the 2012 SportAccord World Mind Games was inaugurated in July 2011 in Beijing.

**Follow-up Question:** What did Lubabalo Kondlo win in the 2012 SportAccord World Mind Games?
**Grounded Answer:** In 2012 he won the silver medal, ... in Beijing, China.

**Follow-up Question:** When and where was the 2012 SportAccord World Mind Games inaugurated?
**Grounded Answer:** The International Mind Sports Association (IMSA) inaugurated the SportAccord World Mind Games December 2011 in Beijing ...

**Prediction:**
Won(Lubabalo Kondlo, a silver medal) is True because In 2012 he won the silver medal at the SportAccord World Mind Games in Beijing, China.
Inaugurated(the 2012 SportAccord World Mind Games, July 2011, Beijing) is False because The International Mind Sports Association (IMSA) inaugurated the SportAccord World Mind Games December 2011 in Beijing.
Won(Lubabalo Kondlo, a silver medal) && Inaugurated(the 2012 SportAccord World Mind Games, July 2011, Beijing) is False.
The claim is *[NOT_SUPPORTED]*.

**Explanation:**
Lubabalo Kondlo won a silver medal in the 2012 SportAccord World Mind Games. However, the event was inaugurated in December 2012, not July 2011, in Beijing.

Table 1: An example from **FOLK** with GPT-3.5 on HoVER, a multi-hop claim verification dataset. We first use LLM to translate the claim into a First-Order-Logic clause (highlighted in orange), consisting of two predicates (highlighted in blue and purple). The LLM then perform knowledge-grounded reasoning to predict label and generate explanation.

## 1 Introduction

Claim verification (Guo et al., 2022) has become increasingly important due to widespread online misinformation (Tian et al., 2023; Jin et al., 2023). Most of the existing claim verification models (Zhou et al., 2019; Jin et al., 2022; Yang et al., 2022; Wadden et al., 2022b; Liu et al., 2020; Zhong et al., 2020) use an automated pipeline that consists of claim detection, evidence retrieval, verdict prediction, and justification production. Despite some early promising results, they rely on the availability of large-scale human-annotated datasets, which pose challenges due to labor-intensive annotation efforts and the need for annotators with specialized domain knowledge. To address the issue of creating large-scale datasets, recent works (Pan et al., 2021; Wright et al., 2022; Lee et al., 2021) focus on claim verification in zero-shot and few-shot scenarios. However, these methods follow the traditional claim verification pipeline, requiring both claim

---

[1] https://github.com/wang2226/FOLK

and annotated evidence for veracity prediction. Additionally, these models often lack proper justifications for their predictions, which are important for human fact-checkers to make the final verdicts. Therefore, we ask the following question: *Can we develop a model capable of performing claim verification without relying on annotated evidence, while generating natural language justifications for its decision-making process?*

To this end, we propose a novel framework First-Order-Logic-Guided Knowledge-Grounded (FOLK) to perform explainable claim verification by leveraging the reasoning capabilities of Large Language Models (LLMs) (Brown et al., 2020; Touvron et al., 2023; Zhang et al., 2022; Chowdhery et al., 2022). Figure 1 illustrates a real-world example from FOLK where it can provide veracity prediction based on logical reasoning and generate an explanation of its decision-making process in a short paragraph. To ensure accurate prediction and provide high-quality explanations, FOLK first translates the input claim into a First-Order-Logic (FOL) (Enderton, 2001) clause consisting of a set of conjunctive predicates. Each predicate represents a part of the claim that needs to be verified. Next, the generated FOL predicates guide LLMs to generate a set of questions and corresponding answers. Although the generated answers may appear coherent and plausible, they often lack factual accuracy due to LLM's hallucination problem (Ji et al., 2023; Ouyang et al., 2022). To address this problem, FOLK controls the knowledge source of the LLMs by grounding the generated answers in real-world truth via retrieving accurate information from trustworthy external knowledge sources (e.g. Google or Wikipedia). Finally, FOLK leverages the reasoning ability of LLMs to evaluate the boolean value of the FOL clause and make a veracity prediction. Given the high stakes involved in claim verification, FOLK prompts the LLMs to generate justifications for their decision-making process in natural language. These justifications are intended to aid human fact-checkers in making the final verdict, enhancing the transparency and interpretability of the model's predictions.

We evaluate our proposed methods on three fact-checking datasets (Jiang et al., 2020; Aly et al., 2021; Wadden et al., 2022a) with the following distinct challenges: multi-hop reasoning, numerical reasoning, combining text and table for reasoning, and open-domain scientific claim verification. Our

experiment results demonstrate that FOLK can verify complex claims while generating explanations to justify its decision-making process. Additionally, we show the effectiveness of FOL-guided claim decomposition and knowledge-grounded reasoning for claim verification.

In summary, our contributions are:

- We introduce a new method to verify claims without the need for annotated evidence.

- We demonstrate the importance of using symbolic language to help claim decomposition and provide knowledge-grounding for LLM to perform reasoning.

- We show that FOLK can generate high-quality explanations to assist human fact-checkers.

## 2 Background

**Claim Verification.** The task of claim verification aims to predict the veracity of a claim by retrieving related evidence documents, selecting the most salient evidence sentences, and predicting the veracity of the claim as *SUPPORTS* or *REFUTES*. Claim verification falls into the broader task of *Fact-checking*, which includes all the steps described in claim verification with the addition of claim detection, a step to determine the check-worthiness of a claim.

While steady progress has been made in this field, recent research focus has shifted to 1) dealing with insufficient evidence (Atanasova et al., 2022) and 2) using explainable fact-checking models to support decision-making (Kotonya and Toni, 2020a). In the line of explainable fact-checking, (Popat et al., 2018) and (Shu et al., 2019) use visualization of neural attention weights as explanations. Although attention-based explanation can provide insights into the deep learning model's decision process, it does not generate human-readable explanations and cannot be interpreted without any prior machine learning knowledge. (Atanasova et al., 2020; Kotonya and Toni, 2020b) formulate the task of generating explanations as extractive summarizing of the ruling comments provided by professional fact-checkers. While their work can generate high-quality explanations based on training data from professional fact-checkers, annotating such datasets is expensive and not feasible at a large scale. Our work explores using reasoning steps as explanations of the model's decision-making

process while generating explanations in natural language.

**Large Language Models for Reasoning.** Large language models have demonstrated strong reasoning abilities through chain-of-thought (CoT) prompting, wherein LLM is prompted to generate its answer following a step-by-step explanation by using just a few examples as prompts. Recent works have shown that CoT prompting can improve performance on reasoning-heavy tasks such as multi-hop question answering, multi-step computation, and common sense reasoning (Nye et al., 2021; Zhou et al., 2022; Kojima et al., 2022).

Verifying complex claims often requires multi-step (multi-hop) reasoning (Mavi et al., 2022), which requires combining information from multiple pieces of evidence to predict the veracity of a claim. Multi-step reasoning can be categorized into forward-reasoning and backward-reasoning (Yu et al., 2023). Forward-reasoning (Creswell et al., 2022; Sanyal et al., 2022; Wei et al., 2022) employs a bottom-up approach that starts with existing knowledge and obtains new knowledge with inference until the goal is met. Backward-reasoning (Min et al., 2019; Press et al., 2022) on the other hand, is goal-driven, which starts from the goal and breaks it down into sub-goals until all of them are solved. Compared to forward reasoning, backward reasoning is more efficient, the divide-and-conquer search scheme can effectively reduce the problem search space. We propose FOLK, a FOL-guided backward reasoning method for claim verification.

Despite the recent progress in using LLMs for reasoning tasks, their capability in verifying claims has not been extensively explored. (Yao et al., 2022) evaluate using LLMs to generate reasoning traces and task-specific actions on fact verification tasks. Their reasoning and action steps are more complex than simple CoT and rely on prompting much larger models (PaLM-540B). Additionally, they test their model's performance on the FEVER dataset (Thorne et al., 2018), which lacks many-hop relations and specialized domain claims. In contrast to their approach, our proposed method demonstrates effectiveness on significantly smaller LLMs without requiring any training, and we test our method on scientific claims.

Contemporaneous to our work, (Peng et al., 2023) propose a set of plug-and-play modules that augment with LLMs to improve the factuality of LLM-generated responses for task-oriented dialogue and question answering. In contrast to their approach, our primary focus is on providing LLMs with knowledge-grounded facts to enable FOL-Guided reasoning for claim verification, rather than solely concentrating on enhancing the factual accuracy of LLMs' responses. ProgramFC (Pan et al., 2023) leverages LLMs to generate computer-program-like functions to guide the reasoning process. In contrast to their approach, which only uses LLMs for claim decomposition, we use LLMs for both claim decomposition and veracity prediction. By using LLMs for veracity prediction, we can not only obtain a comprehensive understanding of LLMs' decision process but also generate explanations for their predictions. Furthermore, ProgramFC is limited to closed-domain, as it needs to first retrieve evidence from a large textual corpus like Wikipedia. FOLK on the other hand, can perform open-domain claim verification since it does not require a pre-defined evidence source.

# 3 Method

Our objective is to predict the veracity of a claim $\mathcal{C}$ without the need for annotated evidence while generating explanations to elucidate the decision-making process of LLMs. As shown in Figure 1, our framework contains three stages. In the *FOL-Guided Claim Decomposition* stage, we first translate the input claim into a FOL clause $\mathcal{P}$, then we use $\mathcal{P}$ to guide LLM to generate a set of intermediate question-answer pairs $(q_i, a_i)$. Each intermediate question $q_i$ represents a specific reasoning step required to verify the claim. In the *Knowledge-Grounding* stage, each $a_i$ represents the answer generated by LLMs that has been verified against ground truth obtained from an external knowledge source. Finally, in the *Veracity Prediction and Explanation Generation* stage, we employ $\mathcal{P}$ to guide the reasoning process of LLMs over the knowledge-grounded question-and-answer pairs. This allows us to make veracity predictions and generate justifications for its underlying reasoning process.

## 3.1 FOL-Guided Claim Decomposition

Although LLMs have displayed decent performance in natural language reasoning tasks, they fall short when asked to directly solve complex reasoning problems. This limitation arises from the lack of systematic generalization capabilities in language models (Valmeekam et al., 2022; Elazar et al., 2021). Recent works have discovered that

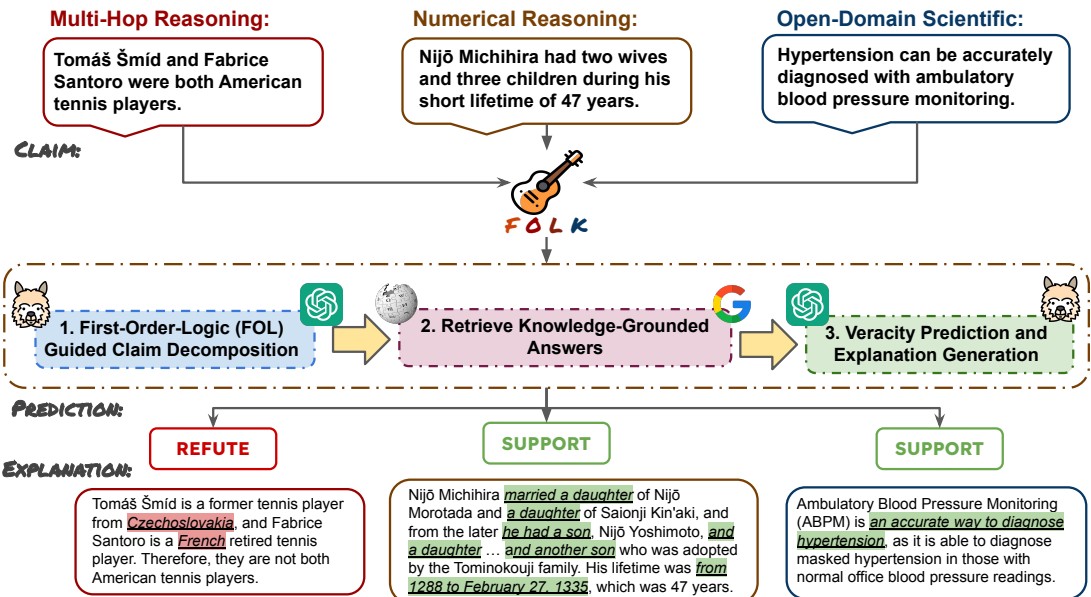

Figure 1: Overview of our *FOLK* framework, which consists of three steps: (i) *FOLK* translates input claim into a FOL clause and uses it to guide LLMs to generate a set of question-and-answer pairs; (ii) *FOLK* then retrieves knowledge-grounded answers from external knowledge-source; and (iii) *FOLK* performs FOL-Guided reasoning over knowledge-grounded answers to make veracity prediction and generate explanations. *FOLK* can perform a variety of reasoning tasks for claim verification, such as multi-hop reasoning, numerical reasoning, and open-domain scientific claim verification.

LLMs are capable of understanding and converting textual input into symbolic languages, such as formal language (Kim, 2021), mathematical equations (He-Yueya et al., 2023), or Python codes (Gao et al., 2022). Inspired by these recent works, we harness the ability of LLMs to translate textual claims into FOL clauses. This allows us to guide LLMs in breaking down claims into various sub-claims.

At this stage, given the input claim $\mathcal{C}$, the LLM first generates a set of predicates $\mathcal{P} = [p_1, ..., p_n]$ that correspond to the sub-claims $\mathcal{C} = [c_1, ..., c_n]$. Each *predicate* $p_i \in P$ is a First-Order Logic (FOL) predicate that guides LLMs to prompt a question-and-answer pair that represents sub-claim $c_i$. The claim $\mathcal{C}$ can be represented as a conjunction of the predicates $\mathcal{C} = p_1 \wedge p_2 \wedge ... \wedge p_n$. To classify the claim $C$ as SUPPORTED, all predicates must evaluate to True. If any of the predicates are False, the claim is classified as REFUTED. By providing the LLMs with symbolic languages such as predicates, alongside a few in-context examples, we observe that LLMs can effectively identify the crucial entities, relations, and facts within the claim. Consequently, LLMs are capable of generating relevant question-and-answer pairs that align with the identified elements.

## 3.2 Retrieve Knowledge-Grounded Answers

Although LLMs exhibit the ability to generate coherent and well-written text, it is worth noting that they can sometimes hallucinate (Ji et al., 2023), and produce text that fails to be grounded in real-world truth. To provide knowledge-grounded answers for the generated intermediate questions, we employ a retriever based on Google Search, via the SerpAPI [2] service. Specifically, we return the top-1 search result returned by Google. While it is important to acknowledge that Google search results may occasionally include inaccurate information, it generally serves as a more reliable source of knowledge compared to the internal knowledge of LLMs. Additionally, in real-world scenarios, when human fact-checkers come across unfamiliar information, they often rely on Google for assistance. Therefore, we consider the answers provided by Google search as knowledge-grounded answers.

## 3.3 Veracity Prediction and Explanation Generation

At this stage, the LLM is asked to make a verdict prediction $\mathcal{V} \in \{\text{SUPPORT}, \text{REFUTE}\}$ and provide an explanation $\mathcal{E}$ to justify its decision.

**Veracity Prediction** Given the input claim $\mathcal{C}$, the

[2] https://serpapi.com/

predicates $[p_1, ..., p_n]$, and knowledge-grounded question-and-answer pairs, FOLK first checks the veracity of each predicate against corresponding knowledge-grounded answers while giving reasons behind its predictions. Once all predicates have been evaluated, FOLK makes a final veracity prediction for the entire clause. In contrast to solely providing LLMs with generated questions and their corresponding grounded answers, we found that the inclusion of predicates assists LLMs in identifying the specific components that require verification, allowing them to offer more targeted explanations. **Explanation Generation** We leverage LLMs' capability to generate coherent language and prompt LLMs to generate a paragraph of human-readable explanation. We evaluate the explanation generated by LLMs with manual evaluation. Furthermore, since claim verification is a high-stake task, it should involve human fact-checkers to make the final decision. Therefore, we provide URL links to the relevant facts, allowing human fact-checkers to reference and validate the information.

## 4 Experiments

We compare FOLK to existing methods on 7 claim verification challenges from three datasets. Our experiment setting is described in Sections 4.1 & 4.2 and we discuss our main results in Section 4.4.

### 4.1 Datasets

We experiment with the challenging datasets listed below. Following existing works (Yoran et al., 2023; Kazemi et al., 2022; Trivedi et al., 2022), to limit the overall experiment costs, we use stratified sampling to select 100 examples from each dataset to ensure a balanced label distribution.

**HoVER** (Jiang et al., 2020) is a multi-hop fact verification dataset created to challenge models to verify complex claims against multiple information sources, or "hop". We use the validation set for evaluation since the test sets are not released publicly. We divide the claims in the validation set based on the number of hops: two-hop claims, three-hop claims, and four-hop claims.

**FEVEROUS** (Aly et al., 2021) is a benchmark dataset for complex claim verification over structured and unstructured data. Each claim is annotated with evidence from sentences and forms in Wikipedia. We selected claims in the validation set with the following challenges to test the effectiveness of our framework: numerical reasoning, multi-

hop reasoning, and combining tables and text.

**SciFact-Open** (Wadden et al., 2022a) is a testing dataset for scientific claim verification. This dataset aims to test existing models' claim verification performance in an open-domain setting. Since the claims in SciFact-Open do not have a global label, we select claims with complete evidence that either support or refute the claim and utilize them as the global label. This dataset tests our model's performance on specialized domains that require domain knowledge to verify.

### 4.2 Baselines

We compare our proposed method against the following four baselines.

**Direct** This baseline simulates using LLM as standalone fact-checkers. We directly ask LLMs to give us veracity predictions and explanations given an input claim, relying solely on LLMs' internal knowledge. It is important to note that we have no control over LLM's knowledge source, and it is possible that LLMs may hallucinate.

**Chain-of-Thought** (Wei et al., 2022) is a popular approach that demonstrates chains of inference to LLMs within an in-context prompt. We decompose the claims by asking LLMs to generate the necessary questions needed to verify the claim. We then prompt LLMs to verify the claims step-by-step given the claims and knowledge-grounded answers.

**Self-Ask** (Press et al., 2022) is a structured prompting approach, where the prompt asks LLMs to decompose complex questions into easier sub-questions that it answers before answering the main question. It is shown to improve the performance of Chain-of-Thought on multi-hop question-answering tasks. We use the same decomposition and knowledge-grounding processes as in CoT. For veracity prediction, we provide both questions and knowledge-grounded answers to LLMs to reason, instead of just the knowledge-grounded answers.

**ProgramFC** (Pan et al., 2023) is a recently proposed baseline for verifying complex claims using LLMs. It contains three settings for knowledge-source: gold-evidence, open-book, and closed-book. To ensure that ProgramFC has the same problem setting as *FOLK*, we use the open-book setting for ProgramFC. Since we only use one reasoning chain, we select N=1 for ProgramFC. Since ProgramFC cannot perform open-domain claim verification, we exclude it from SciFact-Open dataset.

| | HoVER | | | FEVEROUS | | | SciFact-Open |
|---|---|---|---|---|---|---|---|
| | **2-Hop** | **3-Hop** | **4-Hop** | **Numerical** | **Multi-hop** | **Text and Table** | |
| Direct | 57.11 | 44.95 | 55.91 | 48.52 | 50.18 | 59.07 | 49.70 |
| CoT | 53.98 | 46.57 | 47.99 | 49.56 | 60.90 | 61.76 | 63.39 |
| Self-Ask | 54.23 | 48.87 | 51.76 | 55.33 | 61.16 | 54.23 | 60.94 |
| ProgramFC | 71.00 | 51.04 | 52.92 | 54.78 | 59.84 | 51.69 | - |
| **FOLK** | 66.26 | **54.80** | **60.35** | **59.49** | **67.01** | **63.42** | **67.59** |

Table 2: Macro F-1 score of Direct, Chain-of-Thought (CoT), Self-Ask, ProgramFC, and our method *FOLK* on three challenging claim verification datasets. The best results within each dataset are highlighted.

## 4.3 Experiment Settings

The baselines and FOLK use GPT-3.5, *text-davinci-003* (175B) as the underlying LLM. We use *SER-PAPI* as our retrieval engine to obtain knowledge-grounded answers. In addition to the results in Table 2, we perform experiments on smaller LLMs (Touvron et al., 2023): llama-7B, llama-13B, and llama-30B. The results are presented in Table 2. Our prompts are included in B. The number of prompts used varies between 4-6 between the datasets. These prompts are based on random examples from the train and development sets.

## 4.4 Main Results

We report the overall results for FOLK compared to the baselines for claim verification in Table 2. FOLK achieves the best performance on 6 out of 7 evaluation tasks, demonstrating its effectiveness on various reasoning tasks for claim verification. Based on the experiment results, we have the following major observations:

**FOLK is more effective on complex claims.** On HoVER dataset, FOLK outperforms the baselines by 7.37% and 7.94% on three-hop and four-hop claims respectively. This suggests that FOLK becomes more effective on more complex claims as the required reasoning depth increases. Among the baselines, ProgramFC has comparable performance on three-hop claims, which indicates the effectiveness of using symbolic language, such as programming-like language to guide LLMs for claim decomposition for complex claims. However, programming-like language is less effective as claims become more complex. Despite ProgramFC having a performance increase of 3.68% from three-hop to four-hop claims in HoVER, FOLK has a larger performance increase of 10.13%. Suggesting that FOL-guided claim decomposition is more effective on more complex claims.

On FEVEROUS dataset, FOLK outperforms the baselines by 7.52%, 9.57%, and 2.69% on all three tasks respectively. This indicates that FOLK can perform well not only on multi-hop reasoning tasks but also on numerical reasoning and reasoning over text and table.

**FOL-guided Reasoning is more effective than CoT-like Reasoning.** Our FOLK model, which uses FOL-guided decomposition reasoning approach outperforms CoT and Self-Ask baselines on all three datasets. On average, there is an 11.30% improvement. This suggests that FOL-like predicates help LLMs to better decompose claims, and result in more accurate reasoning. This is particularly evident when the claims become more complex: there is a 12.13% improvement in three-hop and a 16.6% improvement in the four-hop setting.

**Knowledge-grouding is more reliable than LLM's internal knowledge.** FOLK exhibits superior performance compared to Direct baseline across all three datasets. This observation indicates the critical role of knowledge-grounding in claim verification, as Direct solely relies on the internal knowledge of LLMs. It is also important to note that the lack of control over the knowledge source in Direct can lead to hallucinations, where LLMs make accurate predictions but for incorrect reasons. For instance, when faced with a claim labeled as SUPPORT, LLMs may correctly predict the outcome despite certain predicates being false.

| | 2-hop | 3-hop | 4-hop |
|---|---|---|---|
| en.wikipedia.org | **66.26** | **54.80** | **60.35** |
| google.com | 62.60 | 50.88 | 54.66 |

Table 3: Ablation study on knowledge source.

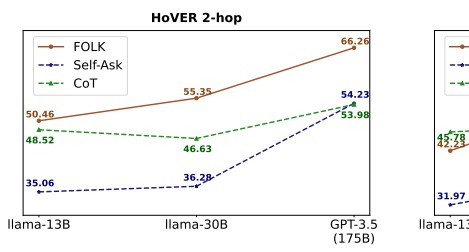 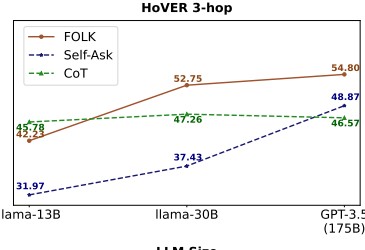 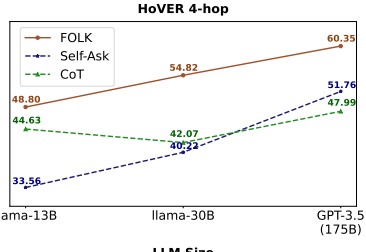

Figure 2: Macro-F1 score of running *FOLK* (brown line), Self-Ask (green dashed line), and CoT (blue dashed line) on HoVER dataset for LLMs with increasing size: llama-13B, llama-30B, and GPT-3.5 (175B).

## 4.5 The Impacts of FOL-Guided Reasoning

To gain more insights on prompting from FOL predicates, we perform an ablation study on the HoVER dataset. The goal is to see whether the performance difference in Table 2 primarily results from FOLK generating better follow-up questions or if the predicates also play a role in constructing the veracity prediction. Specifically, we maintain the CoT prompt format but input knowledge-grounded answers from FOLK. As for Self-Ask, we maintain the Self-Ask prompt format while incorporating follow-up questions generated by FOLK along with their associated knowledge-grounded answers. This guarantees that both CoT and Self-Ask retain their reasoning capabilities while employing identical factual information as provided by FOLK. The results, presented in Table 4, show that FOLK consistently outperforms CoT and Self-Ask in all three tasks. This highlights that the FOL-guided reasoning process enhances the ability of language models to integrate knowledge in multi-hop reasoning scenarios effectively.

## 4.6 The Impacts of Knowledge-Grounding

To better understand the role of knowledge-grounding in LLM's decision process, we perform an ablation study on four multi-hop reasoning tasks. We use the FOLK prompt to generate predicates and decompose the claim, we then compare its performance under two settings. In the first setting, we let LLM reason over the answers it generated itself. In the second setting, we provide LLM

| | 2-hop | 3-hop | 4-hop |
|---|---|---|---|
| CoT using FOLK questions | 57.78 | 41.20 | 44.57 |
| Self-Ask using FOLK questions | 62.00 | 43.25 | 42.86 |
| FOLK | **66.26** | **54.80** | **60.35** |

Table 4: Ablation study on FOL-guided reasoning.

with knowledge-grounded answers. The results are shown in Figure 3, as we can see, FOLK performs better with knowledge-grounded answers. This suggests that by providing knowledge-grounded answers, we can improve LLM's reasoning performance, and alleviate the hallucination problem by providing it facts.

Next, we investigate whether the knowledge source can affect FOLK's performance. Since both HoVER and FEVEROUS datasets are constructed upon Wikipedia pages. We add en.wikipedia.com in front of our query to let it search exclusively from Wikipedia. This is the same way as ProgramFC's open-book setting. We record the performance in Table 3. As we can see, using a more accurate search can lead to better performance.

## 4.7 The Generalization on Different-sized LLMs

To assess whether the performance of FOLK can generalize to smaller LLMs, we compare the performance of FOLK against cot and self-ask on HoVER dataset using two different-sized LLMs: llama-7B and llama-13B. Due to the inability of using ProgramFC prompts to generate programs

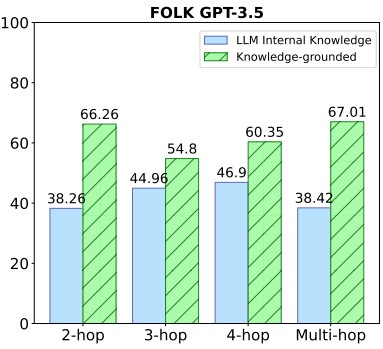

Figure 3: Ablation study on knowledge-grounding for multi-hop reasoning task.

using the llama model, we exclude ProgramFC from our evaluation for this experiment. The results are shown in Figure 2, FOLK can outperform CoT and Self-Ask regardless of the model size, except for 3-hop claims using llama-13B model. As smaller models are less capable for complex reasoning, the performance of Self-Ask decreases significantly with decreasing model size. For CoT, its performance is less sensitive to LLM size compared to Self-Ask. However, these trends are less notable for FOLK. We believe it can attribute to the predicates used to guide LLM to perform high-level reasoning. Our results show that FOLK using llama-30B model can achieve comparable performance to PrgramFC using 5.8x larger GPT-3.5 on three-hop and four-hop claims. This further shows that FOLK is effective on deeper claims and can generalize its performance to smaller LLMs.

## 4.8 Assessing the Quality of Explanations

To measure the quality of the explanations generated by FOLK, we conduct manual evaluations by three annotators. The annotators are graduate students with a background in computer science. Following previous work (Atanasova et al., 2020), we ask annotators to rank explanations generated by CoT, Self-Ask, and FOLK. We choose the following three properties for annotators to rank these explanations:

**Coverage** The explanation can identify and include all salient information and important points that contribute to verifying the claim. We provide fact checkers with annotated gold evidence and ask them whether the generated explanation can effectively encompass the key points present in the gold evidence.

**Soundness** The explanation is logically sound and does not contain any information contradictory to the claim or gold evidence. To prevent annotators from being influenced by the logic generated by FOLK, we do not provide annotators with the predicates generated by FOLK.

**Readability** The explanation is presented in a clear and coherent manner, making it easily understandable. The information is conveyed in a way that is accessible and comprehensible to readers.

We randomly sample 30 instances from the multi-hop reasoning challenge from the FEVEROUS dataset. For each instance, we collect veracity explanations generated by CoT, Self-Ask, and FOLK. During the annotation process, we ask an-

| Annotators | CoT | Self-Ask | FOLK |
|---|---|---|---|
| *Coverage* | | | |
| 1st | 1.90 | 1.95 | 1.75 |
| 2nd | 1.75 | 1.75 | 1.35 |
| 3rd | 1.55 | 1.70 | 1.60 |
| **Avg** | 1.73 | 1.80 | 1.57 |
| *Soundness* | | | |
| 1st | 1.40 | 1.45 | 1.15 |
| 2nd | 1.40 | 1.25 | 1.00 |
| 3rd | 1.05 | 1.05 | 1.05 |
| **Avg** | 1.28 | 1.25 | 1.07 |
| *Readability* | | | |
| 1st | 1.95 | 1.90 | 1.25 |
| 2nd | 1.75 | 1.60 | 1.20 |
| 3rd | 1.35 | 1.50 | 1.35 |
| **Avg** | 1.68 | 1.67 | 1.27 |

Table 5: Mean Average Ranks (MARs) of the explanations for each of the three evaluation criteria. The lower MAR indicates a higher ranking and represents a better quality of an explanation. For each row, the best results from each annotator are underlined, and the best overall results are highlighted in blue.

notators to rank these explanations with the rank 1, 2, and 3 representing first, second, and third place respectively. We also allow ties, meaning that two veracity explanations can receive the same rank if they appear the same. To mitigate potential position bias, we did not provide information about the three different explanations and shuffled them randomly. The annotators worked separately without discussing any details about the annotation task.

**FOLK can generate informative, accurate explanations with great readability.** Table 5 shows the results from the manual evaluation mentioned above. We use Mean Average Ranks (MARs) as our evaluation metrics, where a lower MAR signifies a higher ranking and indicates a better quality of an explanation. To measure the inter-annotator agreement, we compute Krippendorf's $\alpha$ (Hayes and Krippendorff, 2007). The corresponding $\alpha$ values for FOLK are 0.52 for *Coverage*, 0.71 for *Soundness*, and 0.69 for *Readability*, where $\alpha > 0.67$ is considered good agreement. We assume the low agreement on coverage can be attributed to the inherent challenges of ranking tasks for manual evaluation. Small variations in rank positions and annotator bias towards ranking ties may impact the agreement among annotators. We find that explanations generated by FOLK are ranked the best for all criteria, with 0.16 and 0.40 ranking improvements on coverage and readability respec-

tively. While Self-Ask has better prediction results compared to CoT, as shown in Table 2, CoT has a 0.17 MAR improvement compared to Self-Ask. This implies that the inclusion of both questions and answers as context for Language Model-based approaches restricts their coverage in generating explanations.

## 5 Conclusion

In this paper, we propose a novel approach to tackle two major challenges in verifying real-world claims: the scarcity of annotated datasets and the absence of explanations. We introduce FOLK, a reasoning method that leverages First-Order Logic to guide LLMs in decomposing complex claims into sub-claims that can be easily verified through knowledge-grounded reasoning with LLMs.

Our experiment results show that FOLK demonstrates promising performance on three challenging datasets with only 4-6 in-context prompts provided and no additional training. Additionally, we investigate the impact of knowledge grounding and model size on the performance of FOLK. The results indicate that FOLK can make accurate predictions and generate explanations when using a medium-sized LLM such as llama-30B. To evaluate the quality of the explanations generated by FOLK, we conducted manual evaluations by three human annotators. The results of these evaluations demonstrate that FOLK consistently outperforms the baselines in terms of explanation overall quality.

## 6 Limitations

We identify two main limitations of FOLK. First, the claims in our experiments are synthetic and can be decomposed with explicit reasoning based on the claims' syntactic structure. However, real-world claims often possess complex semantic structures, which require implicit reasoning to verify. Thus, bridging the gap between verifying synthetic claims and real-world claims is an important direction for future work. Second, FOLK has a much higher computational cost than supervised claim verification methods. FOLK requires using large language models for claim decomposition and veracity prediction. This results in around $20 per 100 examples using OpenAI API or around 7.5 hours on locally deployed llama-30B models on an 8x A5000 cluster. Therefore, finding ways to infer LLMs more efficiently is urgently needed alongside this research direction.

## 7 Ethical Statement

**Biases.** We acknowledge the possibility of biases existing within the data used for training the language models, as well as in certain factuality assessments. Unfortunately, these factors are beyond our control.

**Intended Use and Misuse Potential.** Our models have the potential to captivate the general public's interest and significantly reduce the workload of human fact-checkers. However, it is essential to recognize that they may also be susceptible to misuse by malicious individuals. Therefore, we strongly urge researchers to approach their utilization with caution and prudence.

**Environmental Impact.** We want to highlight the environmental impact of using large language models, which demand substantial computational costs and rely on GPUs/TPUs for training, which contributes to global warming. However, it is worth noting that our approach does not train such models from scratch. Instead, we use few-shot in-context learning. Nevertheless, the large language models we used in this paper are likely running on GPU(s).

## Acknowledgements

This material is based upon work supported by the Defense Advanced Research Projects Agency (DARPA) under Agreement No. HR0011-22-9-0100, NSF SaTC-2241068, a Cisco Research Award, a Microsoft Accelerate Foundation Models Research Award. The views, opinions and/or findings expressed are those of the author and should not be interpreted as representing the official views or policies of the Department of Defense or the U.S. Government.

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

## A Implementation Details about the Baselines

In this section, we give the implementation details for the three baselines we used in our work.

### A.1 Large Language Models

We use llama models from decapoda-research on Hugging Face. Note this is not the official model weight. Decapoda Research has converted original model weights to work with Transformers. For llama-13B, we load the weight in its original `float 16` precision. Due to limited GPU memory, we load llama-30B in `int 8` precision.

### A.2 ProgramFC

We implement ProgramFC using the code provided by the authors and make necessary changes to accept our custom data input.

# B Prompts

## Listing 1: Chain-of-Thought (CoT) Decompose Prompt

```
Please tell me the necessary questions that need to be answered in order to verify the following claim:

Claim: Howard University Hospital and Providence Hospital are both located in Washington, D.C.
>>>>>>
Followup Question: Where is Howard Hospital located?
Followup Question: Where is Providence Hospital located?
------
Claim: An IndyCar race driver drove a Formula 1 car designed by Peter McCool during the 2007 Formula One season.
>>>>>>
Followup Question: Which Formula 1 car was designed by Peter McCool during the 2007 Formula One season?
Followup Question: Did an IndyCar driver drove a Formula 1 car designed by Peter McCool during the 2007 Formula One season?
------
Claim: Sumo wrestler Toyozakura Toshiaki committed match-fixing, ending his career in 2011 that started in 1989.
>>>>>>
Followup Question: When did Sumo wrestler Toyozakura Toshiaki ended his career?
Followup Question: What is Toyozakura Toshiaki's occupation?
Followup Question: Did Sumo wrestler Toyozakura Toshiaki committed match-fixing?
------
Claim: In 1959, former Chilean boxer Alfredo Cornejo Cuevas (born June 6, 1933) won the gold medal in the welterweight
    division at the Pan American Games (held in Chicago, United States, from August 27 to September 7) in Chicago, United
    States, and the world amateur welterweight title in Mexico City.
>>>>>>
Followup Question: When was Alfredo Cornejo Cuevas born?
Followup Question: Did Alfredo Cornejo Cuevas win the gold metal in the welterweight division at the Pan American Games in
    1959?
Followup Question: Where was The Pan American Games in 1959 held?
Followup Question: Did Alfredo Cornejo Cuevas win the world amateur welterweight title in Mexico City?
------
Claim: %s
>>>>>>
```

## Listing 2: FOLK Decompose Prompt

```
You are given a problem description and a claim. The task is to:
1) define all the predicates in the claim
2) parse the predicates into followup questions
3) answer the followup questions

Claim: Howard University Hospital and Providence Hospital are both located in Washington, D.C.
>>>>>>
Predicates:
Location(Howard Hospital, Washington D.C.) ::: Verify Howard University Hospital is located in Washington, D.C.
Location(Providence Hospital, Washington D.C.) ::: Verify Providence Hospital is located in Washington, D.C.

Followup Question: Where is Howard Hospital located?
Followup Question: Where is Providence Hospital located?
------
Claim: An IndyCar race driver drove a Formula 1 car designed by Peter McCool during the 2007 Formula One season.
>>>>>>
Predicates:
Designed(Peter McCool, a Formula 1 car) ::: Verify a Formula 1 car was designed by Peter McCool during the 2007 Formula One
    season.
Drive(An IndyCar race driver, a Formula 1 car) ::: Verify an IndyCar driver drove a Formula 1 car.

Followup Question: Which Formula 1 car was designed by Peter McCool during the 2007 Formula One season?
Followup Question: Did an IndyCar driver drove a Formula 1 car designed by Peter McCool during the 2007 Formula One season?
------
Claim: Thomas Loren Friedman has won more Pulitzer Prizes than Colson Whitehead
>>>>>>
Predicates:
Won(Thomas Loren Friedman, Pulitzer Prize) ::: Verify the number of Pulitzer Prizes Thomas Loren Friedman has won.
Won(Colson Whitehead, Pulitzer Prize) ::: Verify the number of Pulitzer Prizes Colson Whitehead has won.

Followup Question: How many Pulitzer Prize did Thomas Loren Friedman win?
Followup Question: How many Pulitzer Prize did Colson Whitehead win?
------
Claim: SkyHigh Mount Dandenong (formerly Mount Dandenong Observatory) is a restaurant located on top of Mount Dandenong,
    Victoria, Australia.
>>>>>>
Predicates:
Location(SkyHigh Mount Dandenong, top of Mount Dandenong, Victoria, Australia) ::: Verify that SkyHigh Mount Dandenong is
    located on top of Mount Dandenong, Victoria, Australia.
Known(SkyHigh Mount Dandenong, Mount Dandenong Observatory) ::: Verify that SkyHigh Mount Dandenong is formerly known as Mount
    Dandenong Observatory.

Followup Question: Where is SkyHigh Mount Dandenong located?
Followup Question: Was SkyHigh Mount Dandenong formerly known as Mount Dandenong Observatory?
------
Claim: Shulin, a 33.1288 km (12.7911 sq mi) land located in New Taipei City, China, a country in East Asia, has a total
    population of 183,946 in December 2018.
>>>>>>
Predicates:
Location(Shulin, New Taipei City, Chian) ::: Verify that Shulin is located in New Taipei City, China.
Population(Shulin, 183,946) ::: Verify that Shulin has a total population of 183,946 in December 2018.

Followup Question: Where is Shulin located?
Followup Question: What is the population of Shulin?
------
Claim: Sumo wrestler Toyozakura Toshiaki committed match-fixing, ending his career in 2011 that started in 1989.
>>>>>>
Predicates:
```

```
Ending(Toyozakura Toshiaki, his career in 2011) ::: Verify that Toyozakura Toshiaki ended his career in 2011.
Occupation(Toyozakura Toshiaki, sumo wrestler) ::: Verify that Toyozakura Toshiaki is a sumo wrestler.
Commit(Toyozakura Toshiaki, match-fixing) ::: Verify that Toyozakura Toshiaki committed match-fixing.

Followup Question: When did Sumo wrestler Toyozakura Toshiaki ended his career?
Followup Question: What is Toyozakura Toshiaki's occupation?
Followup Question: Did Sumo wrestler Toyozakura Toshiaki committed match-fixing?
------
Claim: In 1959, former Chilean boxer Alfredo Cornejo Cuevas (born June 6, 1933) won the gold medal in the welterweight
      division at the Pan American Games (held in Chicago, United States, from August 27 to September 7) in Chicago, United
      States, and the world amateur welterweight title in Mexico City.
>>>>>>
Predicates:
Born(Alfredo Cornejo Cuevas, June 6 1933) ::: Verify that Alfredo Cornejo Cuevas was born June 6 1933.
Won(Alfredo Cornejo Cuevas, the gold metal in the welterweight division at the Pan American Games in 1959) ::: Verify that
      Alfredo Cornejo Cuevas won the gold metal in the welterweight division at the Pan American Games in 1959.
Held(The Pan American Games in 1959, Chicago United States) ::: Verify that The Pan American Games in 1959 was held in Chicago
      United States.
Won(Alfredo Cornejo Cuevas, the world amateur welterweight title in Mexico City).

Followup Question: When was Alfredo Cornejo Cuevas born?
Followup Question: Did Alfredo Cornejo Cuevas win the gold metal in the welterweight division at the Pan American Games in
      1959?
Followup Question: Where was The Pan American Games in 1959 held?
Followup Question: Did Alfredo Cornejo Cuevas win the world amateur welterweight title in Mexico City?
------
Claim: The birthplace of American engineer Alfred L.Rives is a plantation near Monticello, the primary residence of Thomas
      Jefferson.
>>>>>>
Predicates:
Birthplace(Alfred L. Rives, a plantation) ::: Verify The birthplace of American engineer Alfred L.Rives is a plantation
Primary residence(Thomas Jefferson, Monticello) ::: Verify Monticello, the primary residence of Thomas Jefferson.
Near(a planation, Monticello) ::: Verify A plantation is near Monticello

Followup Question: Where is the birthplace of Alfred L. Rives?
Followup Question: Where is the primary residence of Thomas Jefferson?
Followup Question: Is the birthplace of Alfred L. Rives near the residence of Thomas Jefferson?
------
Claim: %s
>>>>>>
```

Listing 3: Direct Reasoning Prompt

```
Please verify the following claim and provide explanations:

Claim: The woman the story behind Girl Crazy is credited to is older than Ted Kotcheff.
>>>>>>
This claim is: [NOT_SUPPORTED]
Here are the reasons: The woman behind the story Girl Crazy is Hampton Del Ruth, who was born on September 7, 1879.
Ted Kotcheff was born on April 7, 1931. Hapmpton Del Ruth is not older than Ted Kotcheff.
------
Claim: A hockey team calls the 70,000 capacity Madison Square Garden it's home. That team, along with the New York Islanders,
      and the New Jersey Devils NHL franchise, are popular in the New York metropolitan area.
>>>>>>
This claim is: [NOT_SUPPORTED]
Here are the reasons: Madison Square Garden is the home to New York Rangers and New York Islanders. Both are popular in the
      New York metropolitan area.
Madison Square Garden has a capacity of 19,500, not 70,0000.
------
Claim: The writer of the song Girl Talk and Park So-yeon have both been members of a girl group.
>>>>>>
This claim is: [SUPPORTED]
Here are the reasons: Tionne Watkins is the writer of the song Girl Talk. She was a member of the girl-group TLC.
Park So-yeon is part of a girl group. Therefore, both Tioone Watkins and Park So-yeon have been members of a girl group.
------
Claim: Werner Gunter Jaff\u00e9 Fellner was born in Frankfurt in the German state of Hesse and the fifth-largest city in
      Germany.
>>>>>>
This claim is: [SUPPORTED]
Here are the reasons: Werner Gunter Jaff\u00e9 Fellner was born in Frankfurt.
Frankfurt is in the German state of Hesse and the fifth-largest city in Germany.
------
Claim: %s
>>>>>>
```

Listing 4: CoT Reasoning Prompt

```
Answer the following SUPPORTED / NOT_SUPPORTED questions:

Is it true that The woman the story behind Girl Crazy is credited to is older than Ted Kotcheff. ?
Let's think step by step.

Girl Crazy 's story is credited to Hampton Del Ruth.
Hampton Del Ruth was born on September 7 , 1879.
Ted Kotcheff was born on April 7 , 1931.
>>>>>>
Therefore , the answer is: [NOT_SUPPORTED]
Here are the reasons: The woman behind the story Girl Crazy is Hampton Del Ruth, who was born on September 7, 1879.
Ted Kotcheff was born on April 7, 1931. Hapmpton Del Ruth is not older than Ted Kotcheff.
------
Is it true that A hockey team calls the 70,000 capacity Madison Square Garden it's home. That team, along with the New York
      Islanders, and the New Jersey Devils NHL franchise, are popular in the New York metropolitan area. ?
Let's think step by step.

Madison Square Garden hosts approximately 320 events a year. It is the home to the New York Rangers of the National Hockey
      League.
```

```
Madison Square Garden has a capacity of 19.500.
The New York Islanders are a professional ice hockey team based in Elmont, New York. ...
>>>>>>
Therefore, the answer is: [NOT_SUPPORTED]
Here are the reasons: Madison Square Garden is the home to New York Rangers and New York Islanders. Both are popular in the
    New York metropolitan area.
Madison Square Garden has a capacity of 19,500, not 70,0000.
------
Is it true that The writer of the song Girl Talk and Park So-yeon have both been members of a girl group. ?
Let's think step by step.

Tionne Watkins is the writer of the song Girl Talk.
Park Soyeon is a South Korean singer. She is a former member of the kids girl group I& Girls.
Watkins rose to fame in the early 1990s as a member of the girl-group TLC
>>>>>>
Therefore, the answer is: [SUPPORTED]
Here are the reasons: Tionne Watkins is the writer of the song Girl Talk. She was a member of the girl-group TLC.
Park So-yeon is part of a girl group. Therefore, both Tioone Watkins and Park So-yeon have been members of a girl group.
------
Is it true that Werner Gunter Jaff\u00e9 Fellner was born in Frankfurt in the German state of Hesse and the fifth-largest city
    in Germany. ?
Let's think step by step.

Werner Gunter Jaff\u00e9 Fellner was born in Frankfurt.
Frankfurt is in the German state of Hesse.
Frankfurt is the fifth-largest city in Germany.
>>>>>>
Therefore, the answer is: [SUPPORTED]
Here are the reasons: Werner Gunter Jaff\u00e9 Fellner was born in Frankfurt.
Frankfurt is in the German state of Hesse and the fifth-largest city in Germany.
------
Is it true that %s?
Let's think step by step:

%s
>>>>>>
```

Listing 5: Self-Ask Reasoning Prompt

```
Given a question and a context, provide a SUPPORTED or NOT_SUPPORTED and explain why.

Question:
Is it true that The writer of the song Girl Talk and Park So-yeon have both been members of a girl group. ?

Context:
Who is the writer of the song Girl Talk? Tionne Watkins is the writer of the song Girl Talk.
Is Park So-yeon a member of a girl group? Park Soyeon is a South Korean singer. She is a former member of the kids girl group
    I& Girls.
Is the writer of the song Girl Talk a member of a girl group? Watkins rose to fame in the early 1990s as a member of the girl-
    group TLC
>>>>>>
The claim is [SUPPORTED]. Here are the reasons, Tionne Watkins is the writer of the song Girl Talk, and she fame in the early
    1990s as a member of the girl-group TLC.
Park Soyeon is a South Korean singer. She is a former member of the kids girl group I& Girls.
------
Question:
Is it true that A hockey team calls the 70,000 capacity Madison Square Garden it's home. That team, along with the New York
    Islanders, and the New Jersey Devils NHL franchise, are popular in the New York metropolitan area. ?

Context:
Which hocky team calls Madison Square Garden Home? Madison Square Garden hosts approximately 320 events a year. It is the home
    to the New York Rangers of the National Hockey League
What is the capacity of Madison Square Garden? Madison Square Garden has a capacity of 19.500.
Is New York Islanders popular in New York Metropolitan area? The New York Islanders are a professional ice hockey team based
    in Elmont, New York. ...
>>>>>>
The claim is [NOT_SUPPORTED]. Here are the reasons, Madison Square Garden hosts approximately 320 events a year. It is the
    home to the New York Rangers of the National Hockey League.
and the New York Islanders are a professional ice hockey team based in Elmont, New York. Madison Square Garden has a capacity
    of 19.500, not 70,000.
------
Question:
Is it true that Werner Gunter Jaff\u00e9 Fellner was born in Frankfurt in the German state of Hesse and the fifth-largest city
     in Germany. ?

Context:
Where was Werner Gunter Jaff\u00e9 Fellner born? Werner Gunter Jaff\u00e9 Fellner was born in Frankfurt.
Which state is Frankfurt in? Frankfurt is in the German state of Hesse.
>>>>>>
The claim is [SUPPORTED]. Here are the reasons, Werner Gunter JafFf\u00e9 Fellner was born in Frankfurt and Frankfurt is in
    the German state of Hesse.
------
Question:
Is it true that The American lyricist Tom Jones,  born in 1928, co-authored the screenplay for the musical film The Fantastics
    . ?

Context:
When was Tom Jones born? Thomas Jones Woodward was born in Pontypridd, South Wales, Great Britain on June 7, 1940
What is Tome Jones nationality? Sir Thomas Jones Woodward OBE is a Welsh singer.
Who co-author the musical film The Fantastics? Tome Jones co-authored the musical film The Fantastics.
>>>>>>
The claim is [NOT_SUPPORTED]. Here are the reasons, Sir Thomas Jones Woodward OBE is a Welsh singer and Tome Jones co-authored
        the musical film The Fantastics,
but Thomas Jones Woodward was born in Pontypridd, South Wales, Great Britain on June 7, 1940. Thomas Jones is British, not
    American.
------
Question: Is it true that %s?
```

```
Context:
%s
>>>>>>
```

## Listing 6: FOLK Reasoning Prompt

```
Given a question and a context, provide a [SUPPORTED] or [NOT_SUPPORTED] answer and explain why.

Question:
Is it true that The writer of the song Girl Talk and Park So-yeon have both been members of a girl group.?

Context:
Write(the writer, the song Girl Talk) ::: Verify that the writer of the song Girl Talk
Member(Park So-yeon, a girl group) ::: Verify that Park So-yeon is a memeber of a girl group
Member(the writer, a girl group) ::: Verify that the writer of the song Girl Talk is a member of a gril group

Who is the writer of the song Girl Talk? Tionne Watkins is the writer of the song Girl Talk.
Is Park So-yeon a member of a girl group? Park Soyeon is a South Korean singer. She is a former member of the kids girl group
    I& Girls.
Is the writer of the song Girl Talk a member of a girl group? Watkins rose to fame in the early 1990s as a member of the girl-
    group TLC
>>>>>>
Prediction:
Write(Tionne Watkins, the song Girl Talk) is True because Tionne Watkins is the writer of the song Girl Talk.
Member(Park So-yeon, a girl group) is True because Park Soyeon is a South Korean singer. She is a former member of the kids
    girl group I& Girls.
Member(Tionne Watkins, a girl group) is True because Watkins rose to fame in the early 1990s as a member of the girl-group TLC
Write(Tionne Watkins, the song Girl Talk) && Member(Park So-yeon, a girl group) && Member(Tionne Watkins, a girl group) is
    True.
The claim is [SUPPORTED].

Explanation:
Tionne Watkins, a member of the girl group TLC in the 1990s, is the writer of the song "Girl Talk."
Park Soyeon, a South Korean singer, was formerly part of the girl group I& Girls.
Therefore, both Watkins and Park Soyeon have been members of girl groups in their respective careers.
------
Question:
Is it true that A hockey team calls the 70,000 capacity Madison Square Garden it's home. That team, along with the New York
    Islanders, and the New Jersey Devils NHL franchise, are popular in the New York metropolitan area.?

Context:
Home(a hocky team, Madison Square Garden) ::: Verify that a hockey team calls Madison Square Garden its home.
Capacity(Madison Square Garden, 70,000) ::: Verify that Madison Square Garden has capacity of 70,000.
Popular(New York Islanders, New York Metropolitan area) ::: Verify that New York Islanders are popular in the New York
    metropolitan area.

Which hocky team calls Madison Square Garden Home? Madison Square Garden hosts approximately 320 events a year. It is the home
    to the New York Rangers of the National Hockey League
What is the capacity of Madison Square Garden? Madison Square Garden has a capacity of 19.500.
Is New York Islanders popular in New York Metropolitan area? The New York Islanders are a professional ice hockey team based
    in Elmont, New York. ...
>>>>>>
Prediction:
Home(New York Rangers, Madison Square Garden) is True because Madison Square Garden hosts approximately 320 events a year. It
    is the home to the New York Rangers of the National Hockey League
Capacity(Madison Square Garden, 70,000) is False because Madison Square Garden has a capacity of 19.500.
Popular(New York Islanders, New York Metropolitan area) is True because The New York Islanders are a professional ice hockey
    team based in Elmont, New York. ...
Home(New York Rangers, Madison Square Garden) && Capacity(Madison Square Garden, 70,000) && Popular(New York Islanders, New
    York Metropolitan area) is False.
The claim is [NOT_SUPPORTED].

Explanation:
The New York Rangers, along with the New York Islanders and the New Jersey Devils, are popular National Hockey League (NHL)
    teams in the New York metropolitan area.
Madison Square Garden, a well-known venue in New York City, has a capacity of approximately 19,500, not 70,000.
------
Question:
Is it true that Werner Gunter Jaff\u00e9 Fellner was born in Frankfurt in the German state of Hesse and the fifth-largest city
     in Germany.?

Context:
Born(Werner Gunter Jaff\u00e9 Fellner, Frankfurt)
State(Frankfurt, the German state of Hesse)

Where was Werner Gunter Jaff\u00e9 Fellner born? Werner Gunter Jaff\u00e9 Fellner was born in Frankfurt.
Which state is Frankfurt in? Frankfurt is in the German state of Hesse.
>>>>>>
Prediction:
Born(Werner Gunter Jaff\u00e9 Fellner, Frankfurt) is True because Werner Gunter Jaff\u00e9 Fellner was born in Frankfurt.
State(Frankfurt, the German state of Hesse) is True because Frankfurt is in the German state of Hesse.
Born(Werner Gunter Jaff\u00e9 Fellner, Frankfurt) && State(Frankfurt, the German state of Hesse) is True.
The claim is [SUPPORTED].

Explanation:
Werner Gunter Jaffe Fellner was born in Frankfurt, which is both in the German state of Hesse and the fifth-largest city in
    Germany.
------
Question:
Is it true that The American lyricist Tom Jones,  born in 1928, co-authored the screenplay for the musical film The Fantastics
    .?

Context:
Born(Tom Jones, 1928)
Nationality(Tom Jones, American)
Co-author(Tome Jones, the musical film The Fantastics)

When was Tom Jones born? Thomas Jones Woodward was born in Pontypridd, South Wales, Great Britain on June 7, 1940
```

```
What is Tome Jones nationality? Sir Thomas Jones Woodward OBE is a Welsh singer.
Who co-author the musical film The Fantastics? Tome Jones co-authored the musical film The Fantastics.
>>>>>>
Prediction:
Born(Tom Jones, 1928) is False because Thomas Jones Woodward was born in Pontypridd, South Wales, Great Britain on June 7,
      1940
Nationality(Tom Jones, American) is False because Thomas Jones Woodward is a British singer.
Co-author(Tome Jones, the musical film The Fantastics) is True because Tome Jones co-authored the musical film The Fantastics.
Born(Tom Jones, 1928) && Nationality(Tom Jones, American) && Co-author(Tome Jones, the musical film The Fantastics) is False.
The claim is [NOT_SUPPORTED].

Explanation:
Thomas Jones Woodward was born in Pontypridd, South Wales, Great Britain on June 7, 1940. He is a british singer.
Thomas Jones co-authored the musical film The Fantastics.
------
Question: Is it true that %s?

Context:
%s
>>>>>>
"""
```

# C   Manual Evaluation Example

**Please rank the following three explanations based on the following criteria, ties are allowed:**

- Coverage: The explanation can identify and include all salient information and important points that contribute to verifying the claim.

- Soundness: The explanation is logically sound and does not contain any information contradictory to the claim.

- Readability: The explanation is presented in a clear and coherent manner, making it easily understandable.

```
Claim: Anat Berko was elected to the Knesset as Likud, an alliance of right-wing
parties, had won 30 seats.
```

```
The claim is [SUPPORTED]. Here are the reasons: Anat Berko was elected to the
Knesset as Likud, an alliance of right-wing parties, had won 30 seats in the
November 2015 legislative elections.
```

```
The claim is: [SUPPORTED]. Here are the reasons: Anat Berko was a candidate of
the Likud party. Likud won the 2015 election with 30 seats. Therefore, Anat Berko
was elected to the Knesset as Likud had won 30 seats.
```

```
The claim is [SUPPORTED]. Here are the reasons, Anat Berko was placed 23rd on the
Likud list, and Likud won the 2015 election, winning 30 seats to the Zionist
Union's 24. She was elected to Knesset as Likud had won 30 seats.
```

Coverage: [____________]

Soundness: [____________]

Readability: [____________]

[ Submit ]