# OpenReview forum: "Explainable Claim Verification via Knowledge-Grounded Reasoning with Large Language Models"
_EMNLP/2023/Conference — EMNLP 2023 Findings_

### Official Review · Reviewer_ajs8 · 2023-07-23

**Soundness:** 4

**Excitement:**

3: Ambivalent: It has merits (e.g., it reports state-of-the-art results, the idea is nice), but there are key weaknesses (e.g., it describes incremental work), and it can significantly benefit from another round of revision. However, I won't object to accepting it if my co-reviewers champion it.

**Paper Topic And Main Contributions:**

This paper proposes a method to verify claims using the ability of LLMs to first decompose a given claim into several sub-claims and then verify each sub-claim by retrieving relevant knowledge.

Specifically, they first translate a given claim into a First-Order-Logic clause consisting of several predicates, each corresponding to a part of the claim that needs to be verified.

Then, for each predicate, they create a followup question, retrieve relevant knowledge, and generate an answer for the followup question leveraging the retrieved knowledge.

Then, they evaluate the boolean value of the FOL clause and make a prediction as one of SUPPORTS or REFUTES.

They also prompt the LLM to generate an explanation for its decision-making process.

They evaluate the proposed method on 3 fact-checking datasets and demonstrate its effectiveness.


**Questions For The Authors:**

Not relying on human-annotated data is an advantage on which a a lot of emphasis has been given. However, majority of the recent methods (the ones using LLMs) focus on this setting only. I think a bit extra emphasis has been given to this point.


**Reasons To Accept:**

Claim verification is an important topic with practical significance.

The proposed approach is intuitive, easy to implement, and effective. The experiments show that it leads to some performance improvements.

Furthermore, the paper is well written and provides a good background for the problem statement.


**Reasons To Reject:**

Weaknesses:

The reason behind first creating an FOL predicate for each sub-claim is not very convincing. Since they are not using any logical reasoning tools like CLINGO for solving the predicate. I don’t see a justification for doing this.
Instead, directly instructing the LLMs with a prompt as simple as “Generate questions verifying the sub-claims of the above claim” does very well in directly generating sub-questions that can be used for verifying the claims.

For Chain-of-Thought and Self-Ask baselines, do you provide the retrieved knowledge to verify the sub-claims as done with FOLK? If yes, then how would you justify the better performance of your approach since both these approaches also result in accurate questions for verifying the subclaims.
In no, then verifying each sub-claim (decomposed by existing methods) leveraging the retrieved knowledge should be a fair baseline in comparison to the proposed approach that uses the retrieved knowledge.

Furthermore, since creating FOL predicates is not necessarily a trivial task (atleast more complicated than prompting it to decompose the given claim), an error in creating FOL predicates can propagate in generating the followup questions. On the other hand, directly generating the decomposed sub-claims has been shown to work really well in prior works.


Discrepancies in the presented results of ProgramFC approach. As reported in the ProgramFC paper, it achieves 60.63 on HoVER 3-hop (as compared to 54.80 of FOLK) and similarly for the FEVEROUS dataset. However, this paper reports 51.04 on the HoVER 3-hop. Please provide a justification for this.

A human study on how well the verification process leveraging only the top-1 search result is required. Also, why not 5 or maybe more search results are used as the knowledge because the first result may not always contain sufficient information to verify the sub-claim.

Analyzing the relation between correct claim verification decision and the correct generated explanation can be added in the analysis section.

A lot of emphasis in the paper has been on generating explanations; however, only 30 examples have been evaluated which may not provide enough evidence to support the claims of the paper.

**Reproducibility:**

3: Could reproduce the results with some difficulty. The settings of parameters are underspecified or subjectively determined; the training/evaluation data are not widely available.

**Reviewer Confidence:**

4: Quite sure. I tried to check the important points carefully. It's unlikely, though conceivable, that I missed something that should affect my ratings.

---

> ### Author Rebuttal · Authors · 2023-08-29
>
> We thank the reviewer for the thorough feedback and constructive suggestions.
>
> **Q1:** The reason behind first creating a FOL predicate for each sub-claim is not very convincing. Since they are not using any logical reasoning tools like CLINGO for solving the predicate. I don’t see a justification for doing this. Instead, directly instructing the LLMs with a prompt as simple as “Generate questions verifying the sub-claims of the above claim” does very well in directly generating sub-questions that can be used for verifying the claims.
>
> **A1:** We appreciate you bringing up concern regarding the motivation for using FOL predicate to decompose the claims. We use CoT as one of the baselines, which directly instructs the LLMs with a prompt to generate sub-questions that can be used for verifying the claims. The explicit prompt can be found in *Appendix B*, "Please tell me the necessary questions that need to be answered in order to verify the following claim:". Our experiment shows that FOLK can outperform CoT across all three datasets, which shows the importance of using FOL to guide LLMs for claim decomposition. Additionally, we observe that for more complex claims, FOLK has an improvement of 17.67% and 25.76% over CoT on 3-hop and 4-hop claims respectively. This further shows the importance of using FOL can better decompose more complex claims.
>
> **Q2:** For Chain-of-Thought and Self-Ask baselines, do you provide the retrieved knowledge to verify the sub-claims as done with FOLK? If yes, then how would you justify the better performance of your approach since both these approaches also result in accurate questions for verifying the subclaims. In no, then verifying each sub-claim (decomposed by existing methods) leveraging the retrieved knowledge should be a fair baseline in comparison to the proposed approach that uses the retrieved knowledge.
>
> **A2:** (1) We retrieved knowledge for Chain-of-Thought and Self-Ask baselines, which use the same Serpapi calls as used in FOLK. (2) Although both baselines and FOLK retrieve knowledge from an external source, they each generate decomposed sub-claims differently. FOLK can generate better sub-claims especially when the claim becomes more complex, such as more hops or require numerical reasoning. The decomposed sub-claims by all methods are saved as JSON files in our GitHub repository, which will be made public once the reviewing period ends.
>
> **Q3:** Furthermore, since creating FOL predicates is not necessarily a trivial task (at least more complicated than prompting it to decompose the given claim), an error in creating FOL predicates can propagate in generating the followup questions. On the other hand, directly generating the decomposed sub-claims has been shown to work really well in prior works.
>
> **A3:** Thank you for bringing up the concern of error propagation of creating FOL predicates. This is a valid concern that could happen. Since checking the correctness of generated FOL predicates is a non-trivial task, we would like to explore this research direction further.
> Our experiment shows that FOLK can outperform methods that directly generate the decomposed sub-claims (CoT and Self-Ask).
>
> **Q4:** Discrepancies in the presented results of ProgramFC approach. As reported in the ProgramFC paper, it achieves 60.63 on HoVER 3-hop (as compared to 54.80 of FOLK) and similarly for the FEVEROUS dataset. However, this paper reports 51.04 on the HoVER 3-hop. Please provide a justification for this.
>
> **A4:** As stated in Section 4.1, we use stratified sampling to select 100 examples for each sub-dataset (e.g. 3-hop, 4-hop, numerical) in HoVER and FEVEROUS datasets. This is to limit the overall API call cost, which is a common practice used by other works [1,2] that require OpenAI API calls. On the other hand, ProgramFC performs experiments on the entire validation set of HoVER and FEVEROUS. Therefore, the performance of ProgramFC is different from our paper.
>
> [1] Yoran, Ori, et al. "Answering questions by meta-reasoning over multiple chains of thought." arXiv preprint arXiv:2304.13007 (2023).
>
> [2] Trivedi, Harsh, et al. "Interleaving retrieval with chain-of-thought reasoning for knowledge-intensive multi-step questions." arXiv preprint arXiv:2212.10509 (2022).
>
> **Q5:** A human study on how well the verification process leveraging only the top-1 search result is required. Also, why not 5 or maybe more search results are used as the knowledge because the first result may not always contain sufficient information to verify the sub-claim.
>
> **A5:** (1) We appreciate your suggestion on evaluating the quality of search results. We will add its study results in the next version of the paper. (2) Adding the top 5 search results could potentially add more information. However, it also adds to the API cost. Additionally, we found that the source of knowledge has a significant impact on the performance, as shown in Table 3.
>
> **Q6:** Analyzing the relation between correct claim verification decision and the correct generated explanation can be added in the analysis section.
>
> **A6:** We appreciate your suggestion on analyzing the relation between correct verification decision and correct generated explanation. Since performing human evaluation takes time to facilitate, we will add the results in the next version of our paper.
>
> **Q7:** A lot of emphasis in the paper has been on generating explanations; however, only 30 examples have been evaluated which may not provide enough evidence to support the claims of the paper.
>
> **A7:** We follow the practice from similar works [1,2] on human evaluation of generated explanations. These two works use 40 instances for human evaluation.
>
> [1] Atanasova, Pepa, et al. "Generating fact checking explanations." ACL (2020).
>
> [2] Jolly, Shailza, Pepa Atanasova, and Isabelle Augenstein. "Generating fluent fact checking explanations with unsupervised post-editing." arXiv (2021).
>
> **Q8:** Not relying on human-annotated data is an advantage on which a a lot of emphasis has been given. However, majority of the recent methods (the ones using LLMs) focus on this setting only. I think a bit extra emphasis has been given to this point.
>
> **A8:** While methods that use LLMs can verify claims without human-annotated data, our method adds better accuracy and interpretability, which are two important areas to consider for the claim verification task.

---

### Official Review · Reviewer_UJsT · 2023-08-02

**Soundness:** 2

**Excitement:**

3: Ambivalent: It has merits (e.g., it reports state-of-the-art results, the idea is nice), but there are key weaknesses (e.g., it describes incremental work), and it can significantly benefit from another round of revision. However, I won't object to accepting it if my co-reviewers champion it.

**Missing References:**

For the "Self-Ask" baseline, I think you also need to read this paper "LEAST-TO-MOST PROMPTING ENABLES COMPLEX REASONING IN LARGE LANGUAGE MODELS"

**Paper Topic And Main Contributions:**

The article discusses the importance of claim verification in combating misinformation and the challenge of verifying claims without relying on expensive human-annotated data. The article proposes a new approach called First-Order-Logic-Guided Knowledge-Grounded (FOLK) Reasoning. This method can verify complex claims and generate explanations without annotated evidence using Large Language Models (LLMs). FOLK translates the claim into a First-Order-Logic (FOL) clause and performs FOL-guided reasoning over knowledge-grounded question-and-answer pairs to make veracity predictions and generate explanations to justify its decision-making process. The results of experiments indicate that FOLK outperforms strong baselines on three datasets encompassing various claim verification challenges. The code and data are available for others to use.

**Reasons To Accept:**

1. The method proposed by the authors is interesting.
2. Experiments on three datasets (HoVER, FEVEROUS, and SciFact-Open) shows the effectiveness of this paper.

**Reasons To Reject:**

1. What about the efficiency of this method? Does calling a search engine cost much more time?
2. The baseline of this paper is unfairly compared. Most baselines do not incorporate a search engine and consequently are not conducted in an open book setting, including "Direct," "CoT," and "Self-Ask." In contrast, the prompt method proposed by the authors uses a search engine to get nearly perfect knowledge. Such a comparison makes the experimental results highly doubtful.  In my understanding, tool-using works such as Toolformer[1] are compatible with CoT and Self-Ask. The authors should add retrieved knowledge to those methods too. Moreover, the authors should split the main result in Table 2 into more tables with different settings, such as one table for models in the open book setting and one for the closed setting.
3. The ablation study is weak. Authors only show the performance when Knowledge Grounding (search engine) is removed. However, I am curious why authors must use the first-order-logic form of atomic claims that must be checked. Can you use a free text form, like what ToolFormer[1] does, to call API? Can using the first-order logic give more performance improvement?
4. The motivations in the Abstract and Introduction don't make sense. There are two motivations why the authors design this method: 1) how to verify claims without relying on human-annotated data is still unresolved 2) previous models cannot provide comprehensive explanations that can justify their decisions. I don't see how the prompt method solves these two problems. Authors should notice that it is the LLMs themselves who solve these two questions, not the prompt method in this paper. The in-context learning or zero-shot ability is emergent when we scale up LMs. The ability to generate fluent and logically correct explanations emerges when we scale up the LMs.
5. A writing tip: you need to explain what "Followup Question" and "Grounded Answer" are in Figure 1. It is better to be self-contained.

Ref:
[1] Schick, Timo, et al. "Toolformer: Language models can teach themselves to use tools." arXiv preprint arXiv:2302.04761 (2023).

**Reproducibility:**

3: Could reproduce the results with some difficulty. The settings of parameters are underspecified or subjectively determined; the training/evaluation data are not widely available.

**Reviewer Confidence:**

4: Quite sure. I tried to check the important points carefully. It's unlikely, though conceivable, that I missed something that should affect my ratings.

---

> ### Author Rebuttal · Authors · 2023-08-29
>
> We thank the reviewer for the thorough feedback and constructive suggestions.
>
> **Q1:** The baseline of this paper is unfairly compared. Most baselines do not incorporate a search engine and consequently are not conducted in an open book setting, including "Direct," "CoT," and "Self-Ask."
>
> **A1:** As stated in Section 4.2, there are four baselines used in this paper: Direct, CoT, Self-Ask, and ProgramFC. All baselines *except for Direct* retrieve knowledge from external sources, instead of purely relying on LLMs' internal knowledge. The decomposed sub-claims by all methods are saved as JSON files in our GitHub repository, which will be made public once the reviewing period ends. The table below shows the knowledge source for each baseline:
>
> | *Baseline* | *Knowledge Source* |
> | ------ | ------ |
> | Direct | LLMs' Internal Knowledge |
> | **CoT** | **Retrieved from Serpapi (same as FOLK)** |
> | **Self-Ask** | **Retrieved from Serpapi (same as FOLK)** |
> | ProgramFC | Wikipedia corpus (Open-book setting in ProgramFC) |
>
> We select Direct baseline as a vanilla method for directly using LLMs for claim verification tasks. As stated in section 4.2, all the other baselines retrieve external knowledge. We will make it more clear on the knowledge source used by CoT and Self-Ask in the next version.
>
> **Q2:** The motivations in the Abstract and Introduction don't make sense. There are two motivations why the authors designed this method: 1) how to verify claims without relying on human-annotated data is still unresolved 2) previous models cannot provide comprehensive explanations that can justify their decisions. I don't see how the prompt method solves these two problems. Authors should notice that it is the LLMs themselves who solve these two questions, not the prompt method in this paper. The in-context learning or zero-shot ability is emergent when we scale up LMs. The ability to generate fluent and logically correct explanations emerges when we scale up the LMs.
>
> **A2:** Our experiments have shown that LLMs themselves cannot solve the two questions: 1) Table 2 shows that when testing on the same backbone LLMs, our proposed method can achieve better performance than CoT and Self-Ask, which shows the effectiveness of our proposed method. 2) LLMs themselves can generate fluent and logically correct text. However, they can suffer from hallucination problems and produce fluent text with non-factual information. Figure 3 shows the importance of retrieving knowledge from reliable external sources can help with this issue.
>
> **Q3:** Moreover, the authors should split the main result in Table 2 into more tables with different settings, such as one table for models in the open book setting and one for the closed setting.
>
> **A3:** Thank you for your suggestions. Due to the 8-page limit, we cannot split Table 2 into more tables. However, we will consider doing so in our next version.
>
> **Q4:** The ablation study is weak. Authors only show the performance when Knowledge Grounding (search engine) is removed.
>
> **A4:** Figure 3 shows FOLK's performance with and without knowledge-grounding. Additionally, Table 3 shows the performance of using knowledge from different sources.
>
> **Q5:** However, I am curious why authors must use the first-order-logic form of atomic claims that must be checked. Can you use a free text form, like what ToolFormer[1] does, to call API? Can using the first-order logic give more performance improvement?
>
> **A5:** We use FOL predicates to decompose better sub-claims, which is crucial when the input claims are complex or require explicit reasoning abilities, such as numerical reasoning. As shown in Table 2, FOLK achieves better performance than methods without using FOL predicates for claim decomposition. We can use ToolFormer's free-form style to call APIs. However, it will still need FOL predicates to help decompose complex claims.
>
> **Q6:** A writing tip: you need to explain what "Followup Question" and "Grounded Answer" are in Figure 1. It is better to be self-contained.
>
> **A6:** Thank you for your suggestion. We will add these two terms in the caption of Figure 1 to make it self-contained.
>
> **Q7:** For the "Self-Ask" baseline, I think you also need to read this paper "LEAST-TO-MOST PROMPTING ENABLES COMPLEX REASONING IN LARGE LANGUAGE MODELS"
>
> **A7:** Thank you for your suggestion. The key idea of Least-to-Most prompting strategy is to let LLM answer subproblems sequentially. However, for fact-checking, the sub-claims are often not related in a sequential manner and cannot be answered by using sub-answers from previous sub-questions.
>
> **Q8:** What about the efficiency of this method? Does calling a search engine cost much more time?
>
> **A8:** Calling Serpapi costs around 0.5 seconds per request. Additionally, this service allows for high throughput. This method has been adopted as an efficient way to retrieve knowledge from external sources by other works [1,2,3].
>
> [1] Yoran, Ori, et al. "Answering questions by meta-reasoning over multiple chains of thought." arXiv preprint arXiv:2304.13007 (2023).
>
> [2] Press, Ofir, et al. "Measuring and narrowing the compositionality gap in language models." arXiv preprint arXiv:2210.03350 (2022).
>
> [3] Lazaridou, Angeliki, et al. "Internet-augmented language models through few-shot prompting for open-domain question answering." arXiv preprint arXiv:2203.05115 (2022).

---

### Official Review · Reviewer_6KM3 · 2023-08-05

**Soundness:** 4

**Excitement:**

3: Ambivalent: It has merits (e.g., it reports state-of-the-art results, the idea is nice), but there are key weaknesses (e.g., it describes incremental work), and it can significantly benefit from another round of revision. However, I won't object to accepting it if my co-reviewers champion it.

**Paper Topic And Main Contributions:**

The paper evaluates GPT-3.5 (text-davinci-003) on 3 datasets (HoVER, FEVEROUS, and SciFact-Open), by generating intermediate questions to verify a claim (like "Is it true that The writer of the song Girl Talk and Park So-yeon have both been members of a girl group.?") which are directly answered by Google (through serpapi.com) which are then used to produce an answer and an explanation.

Different prompt formats are compared. One baseline is "CoT" which in first generates subquestions (e.g., "Who is the writer of the song Girl Talk?"), the answers are gathered from Google (e.g., "Tionne Watkins is the writer of the song Girl Talk") and listed as the CoT steps. After the steps, the answer is given, followed by an "explanation" (a bit weird that the explanation comes after the answer). The similar "Self-Ask" format shows both questions + answers as the steps.

The paper's main contribution is an improvement over this CoT baseline, called FOLK, which is to first find prompt for the first-order-logic-style predicates in the claim (e.g., the somewhat weird "Write(the writer, the song Girl Talk) ::: Verify that the writer of the song Girl Talk"), followed by a list of follow-up questions implied by the predicates. These questions are answered by Google, and then the final prompt utilized a context of predicates and follow-up questions + answers.

The results indicate that the extra step of prompting for FOL predicates is improving performance, sometimes quite substantially (e.g., 4-hop HoVER questions go from 48 to 60 macro-F1).

Additional analysis includes evaluation also on Llama-7b and -13b (they're weaker, but FOLK tends to be better across the board), and performance when intermediate questions are directly answered by LLM rather than by Google (substantially worse).

There's also a human evaluation of the explanation quality, and FOLK explanation is ranking higher than the compared prompts.

All the explicit prompts are provided in the appendix which is very helpful.

**Reasons To Accept:**

Introduces a simple, but potentially effective, version of LLM prompting which can be of practical use. Evidence for how LLMs can be helped by forcing it to "think" in more structured terms, such as FOL predicates.

Shows the effectiveness of generating simple enough intermediate questions that can be directly answered by a Google search.

**Reasons To Reject:**

It's not clear to what extent this is "just" a prompt engineering exercise vs more profound insights from prompting for FOL predicates. One possible additional experiment might be to use the follow-up questions from FOLK in the self-ask (or CoT) prompts, to see if the performance difference is mostly from FOLK generating better follow-up questions or the predicates also playing a role in constructing the answer.

**Reproducibility:**

4: Could mostly reproduce the results, but there may be some variation because of sample variance or minor variations in their interpretation of the protocol or method.

**Reviewer Confidence:**

4: Quite sure. I tried to check the important points carefully. It's unlikely, though conceivable, that I missed something that should affect my ratings.

---

> ### Author Rebuttal · Authors · 2023-08-29
>
> We thank the reviewer for the thorough feedback and constructive suggestions.
>
> **Q1:** It's not clear to what extent this is "just" a prompt engineering exercise vs more profound insights from prompting for FOL predicates.
>
> **A1:** Our paper aims to solve two important challenges within the claim verification task: 1) Creating human-annotated data for claim verification is expensive and domain-specific. Our proposed method retrieves evidence from external sources and leverages LLMs' reasoning ability to verify claims without relying on annotated evidence. 2) It is important for models to provide reliable explanations that can justify their decisions and assist human fact-checkers. Our proposed method aims to solve these two challenges specifically by leveraging LLMs and retrieving from external knowledge sources.
>
> **Q2:** One possible additional experiment might be to use the follow-up questions from FOLK in the self-ask (or CoT) prompts, to see if the performance difference is mostly from FOLK generating better follow-up questions or the predicates also playing a role in constructing the answer.
>
> **A2:** Thank you for your suggestion. Following your recommendation, we conduct experiments using FOLK-generated follow-up questions in CoT and Self-Ask prompts to derive veracity prediction. We report the experiment results in the following table:
>
> | *Methods* | *2-hop* | *3-hop* | *4-hop* |
> | :------ | :------: | :------: | :------: |
> | CoT | 53.98 | 46.57 | 47.99 |
> | **CoT using FOLK Followup Questions** | **57.78** | **41.20** | **44.57** |
> | Self-Ask | 54.23 | 48.87 | 51.76 |
> | **Self-Ask using FOLK Followup Questions** | **62.00** | **43.25** | **42.86** |
> | FOLK | 66.26 | 54.80 | 60.35 |
>
> As shown in the table above, both CoT and Self-Ask achieve better performance on 2-hop claims using FOLK-generated questions, but not on 3-hop and 4-hop claims. This shows that having explicit predicates in the prediction prompt helps LLMs' reasoning process.

---

### Meta-Review · Area_Chair_MHMZ · 2023-09-20

**Recommendation:** 3

**Metareview:**

Summary: This paper proposes to verify claims by generating intermediate questions. Google's search results are used to answer these questions, and the answers are then used to produce an answer and an explanation. The paper compares different prompts, such as CoT and Self-Ask. The main contribution of the paper is an improved prompt format called FOLK, which involves finding prompts for first-order-logic-style predicates in the claim and generating follow-up questions based on those predicates. The results show that the FOLK approach improves performance, especially for more complex questions.

Strength: This paper presents a straightforward yet promising prompting method for fact-checking. The paper is well-written and easy to follow. Additional experimental results in the author's response further indicate that the presence of the FOL forms helps LLMs combine knowledge in multi-hop cases. Most of the reviewers agree that this is a solid contribution but the excitement is relatively limited.

Weaknesses: I don't think there are any major weaknesses -- most of the weaknesses have been addressed during the discussion phase. The authors should take the experimental results during rebuttal into account when preparing for the final version of the paper.

---

### Decision · Program_Chairs · 2023-10-07

**Decision:**

Accept-Findings

**Comment:**

Summary: This paper proposes to verify claims by generating intermediate questions. Google's search results are used to answer these questions, and the answers are then used to produce an answer and an explanation. The paper compares different prompts, such as CoT and Self-Ask. The main contribution of the paper is an improved prompt format called FOLK, which involves finding prompts for first-order-logic-style predicates in the claim and generating follow-up questions based on those predicates. The results show that the FOLK approach improves performance, especially for more complex questions.

Strength: This paper presents a straightforward yet promising prompting method for fact-checking. The paper is well-written and easy to follow. Additional experimental results in the author's response further indicate that the presence of the FOL forms helps LLMs combine knowledge in multi-hop cases. Most of the reviewers agree that this is a solid contribution but the excitement is relatively limited.

Weaknesses: I don't think there are any major weaknesses -- most of the weaknesses have been addressed during the discussion phase. The authors should take the experimental results during rebuttal into account when preparing for the final version of the paper.